# Hierarchical Autoencoder-based Lossy Compression for Large-scale High-resolution Scientific Data

## Abstract

Lossy compression has become an important technique to reduce data size in many domains. This type of compression is especially valuable for large-scale scientific data, whose size ranges up to several petabytes. Although Autoencoder-based models have been successfully leveraged to compress images and videos, such neural networks have not widely gained attention in the scientific data domain. Our work presents a neural network that not only significantly compresses large-scale scientific data but also maintains high reconstruction quality. The proposed model is tested with scientific benchmark data available publicly and applied to a large-scale high-resolution climate modeling data set. Our model achieves a compression ratio of 140 on several benchmark data sets without compromising the reconstruction quality. 2D simulation data from the High-Resolution Community Earth System Model (CESM) Version 1.3 over 500 years are also being compressed with a compression ratio of 200 while the reconstruction error is negligible for scientific analysis.

## 1 Introduction

Over the past few decades, the amount of information available for analysis has increased significantly. Scientific instruments and related computation systems such as Linac Coherent Light Source (Laboratory, 2023), the Very Large Array Radio Telescope (Observatory, 2023) and high-resolution climate modeling (Chang et al., 2020), produce a massive amount of data and put a huge burden on the existing storage system. It is important to design efficient compression models that are able to reduce the data size for storage while maintaining the key information for analysis.

Data compression can be lossless and lossy. Lossless compression, whose reconstruction is exactly the same as the original data, suffers from a low compression ratio (around 2:1 (Son et al., 2014)) on floating-point datasets (Deutsch, 1996; Collet & Kucherawy, 2015). Meanwhile, lossy compression removes imperceptible details to achieve a much higher compression ratio. Despite the loss of information, the quality of data reconstructed by lossy compression schemes is generally acceptable and usable (Wallace, 1991). The nature of lossy compression has driven scientists and engineers to implement many compression algorithms and methods to substantially reduce the size of scientific data (Di & Cappello, 2016; Tao et al., 2017), whose size is often enormous (might be up to 32 exabytes (Kim et al., 2018)). Furthermore, recent studies by (Baker et al., 2014), (Sasaki et al., 2015), and (Baker et al., 2017) showed that reconstruction data from lossy compression can be used for post-hoc analyses.

In recent years, both scientific and engineering communities have focused on developing neural network models for computer vision (Tan & Le, 2021), natural language processing (Vaswani et al., 2017; Devlin et al., 2018), and compression (Johnston et al., 2019). Among numerous types of deep learning models, Autoencoder (AE) has gained tremendous attention because of its capability to learn data representation. AE is a type of neural network that can efficiently learn the representation of input data in an unsupervised manner for reconstruction. Internally, the network contains a bottleneck layer, whose representation is much smaller than its inputs in terms of size. Therefore, AE is primarily used for dimension reduction and feature extraction. Many variations of AE have been developed to improve the quality of reconstructed data (Kingma & Welling, 2013; Vahdat & Kautz, 2020). Although AE has been shown to be successful in lossy image and

video compression (Minnen & Singh, 2020), there are only a few number of studies leveraging this type of neural network for scientific data compression (Liu et al., 2021a).

In this work, we explore the possibility of leveraging a lossy AE-based compression model to compress scientific data. Specifically, this work aims to achieve high reconstruction quality of data at a very low bit rate, below 0.50. We propose a AE-based model that is capable of significantly reducing data size without compromising data quality. The key contributions of this work are as follows:

- Targeting a very low bit rate region, we implement our own architecture to significantly compress our simulation data from high-resolution HR-CESM1.3 data sets. Benchmark results are also presented to ensure the performance of our model.

- We incorporate masking layers and several preprocessing techniques to significantly improve the compression performance of our model.

## 2 Related Work

Traditional lossy compression for scientific data could be categorized into two types: prediction-based and transform-based. Transform compression (e.g. ZFP (Lindstrom, 2014)) transformed the data before applying other techniques, e.g. embedded coding, to truncate the transformed data. Coefficients and bit-planes determined by the model were used to decompress data. Increasing the number of coefficients and bit-planes improved the quality of reconstructed data but decreased the compression ratio.

On the other hand, prediction-based models, such as SZ (Di & Cappello, 2016; Tao et al., 2017), FPZIP (Lindstrom & Isenburg, 2006), predicted the target data using previous reconstructed data points (Di & Cappello, 2016; Tao et al., 2017). Similar to transform-based models, the authors found that the fidelity of the reconstructed data degraded when a high compression ratio was required. Prediction-based models have been shown to have high reconstruction quality at a high compression ratio, which leads to more studies to improve the performance of this type of compression (Zhao et al., 2021).

Recently, deep learning models have been leveraged to compress many types of data. Many AE-based models showed remarkable results in image and volumetric compression tasks.

Balle et al. (Ballé et al., 2016) introduced an effective end-to-end AE-based model to compress images. The authors trained their models to optimize rate-distortion performance. In order to balance the trade-off between the quality of reconstructed data and compression ratio, both losses for reconstruction and compression rate were minimized simultaneously. Since the quantization layer of their compression models prevented the gradients from flowing through the networks, independently and identically distributed uniform noise was used to replace the quantization layer during training. The added noise enabled the back-propagation without significantly deteriorating the performance of the quantization layer when compressing images.

Models with two levels of quantization were also investigated in (Ballé et al., 2018b). The second layer not only provided fine-grained quantization but also acted as a prior for the first quantization layer. Moreover, arithmetic encoding (Rissanen, 1976) (Rissanen & Langdon, 1979) was implemented instead of variants of Huffman coding (Huffman, 1952). Integer quantization, proposed by (Ballé et al., 2018a), was applied to quantization layers to eliminate the dependence on hardware-platform floating-point implementation, which varied from machine to machine, during compression.

Adopting the idea of two-level quantization, several studies have been conducted to improve the capability of neural networks in image compression. Minnen et al. (Minnen et al., 2018) built an autoregressive model. The first quantization layer, which received inputs from the prior given by the second quantization and from the encoder, autoregressively processes data representation to produce high quality images. Their neural networks were also among the first machine learning models that outperformed the state-of-the-art BPG compression (Bellard, 2023). However, autoregression by its nature prevented neural networks to compute in parallel. Models created by (Minnen & Singh, 2020) eliminated the autoregressive layer and replaced it with multiple splitting layers, which enabled the decoder to comprehensively learn different sets of channels in

parallel. Additionally, optimization for compression speed using neural networks was addressed by (Johnston et al., 2019), which suggested several methods to improve compression performance.

Compression on audio signals using AE-based neural networks has also experienced much progress. The work of (Kim et al., 2022) outperformed MP3 in both compression ratio and audio quality. Their models adopted vector quantization techniques proposed by (Van Den Oord et al., 2017). The authors not only optimized signal losses in the time domain but also minimized reconstruction losses in the frequency domain. Furthermore, the coupling of AE and Generative Adversarial Networks (GAN) (Goodfellow et al., 2020) was leveraged to achieve a high-quality compression model.

Neural networks have also been implemented to compress volumetric scene data. Kim et al. (Kim et al., 2022) replaced fine-grain compression layers in their tree-based models with neural networks, which greatly enhanced the performance on volumetric representation. Coordinate networks by (Martel et al., 2021) not only focused on learning the scene representation but also provided great compression capability.

However, image and video compression models mainly reconstructed integer pixels (or voxels), which were only a subset of scientific data, where data types ranged from integer to floating-point. As a result, several studies using neural networks to enhance scientific data compression have been conducted. Glaws et al. (Glaws et al., 2020) proposed an AE model, which was built upon 12 residual blocks of convolution layers. The authors also incorporated three compression layers to reduce the dimensions of the data in their AE's architecture. The model was trained to compress turbulence data with a fixed compression ratio of 64.

Liu et al. (Liu et al., 2021b) introduced a seven-layer AE model to compress scientific data. The encoder was comprised of three layers of fully connected layers, each of which compressed input data by eight folds. Theoretically, the encoder could compress data by 512x ($8^3$). Similar to the encoder, the decoder had three fully connected layers to decompress the compressed data. Between the encoder and decoder, a bottleneck contained latent variables, whose size was much smaller than the inputs. However, this work mainly focused on small-scale 1D data, whereas our models learned data representation in higher dimensions, particularly in 2D and 3D. Another limitation of this model was that only CPUs were used for compression, which did not fully utilize the parallel computing power offered by GPUs (Nasari et al., 2022).

Recently, a compression method proposed by Liu et al.(Liu et al., 2021a) achieved great results for 2D and 3D data. Their AESZ framework was comprised of a Lorenzo prediction model and an AE model, each of which compressed data independently. Compression outcomes from both models were then compared for the framework to select the model for the data being compressed. The compression ratio of their proposed framework on many scientific data sets surpassed results from other hand-engineered models, e.g. and other AE-based models. However, instead of optimizing one particular model for each input, the framework employed two distinct models to compress the same data.

## 3 Methods

Our proposed model is built upon three main components: an encoder network (E), a quantizer (Q), and a decoder network (D). The encoder network encodes data inputs to output latent variables $z_e$. The quantizer then processes $z_e$ to produce a quantized latent representation $z_q$. Finally, the decoder network reconstructs data from the compressed representation $z_q$ to output $\hat{x}$. The whole model is trained in an end-to-end fashion to minimize a reconstruction loss and constraint losses imposed by codebooks of the quantizer. The model architecture is depicted in Figure 1.

The detailed implementation of the model is present in Table 1. Each stage (EncRes) of the Encoder is connected to an intermediate convolution layer. The intermediate layer acts as a bridge to map the number of channels to the desired vector dimension of the quantization layer. The output representation is then quantized using the corresponding codebook.

### 3.1 Encoder & Decoder Architecture

As mentioned above, the encoder is trained to extract data representation into latent spaces, whereas the decoder decodes the latent variables to reconstruct the given data. There are two most widely used re-

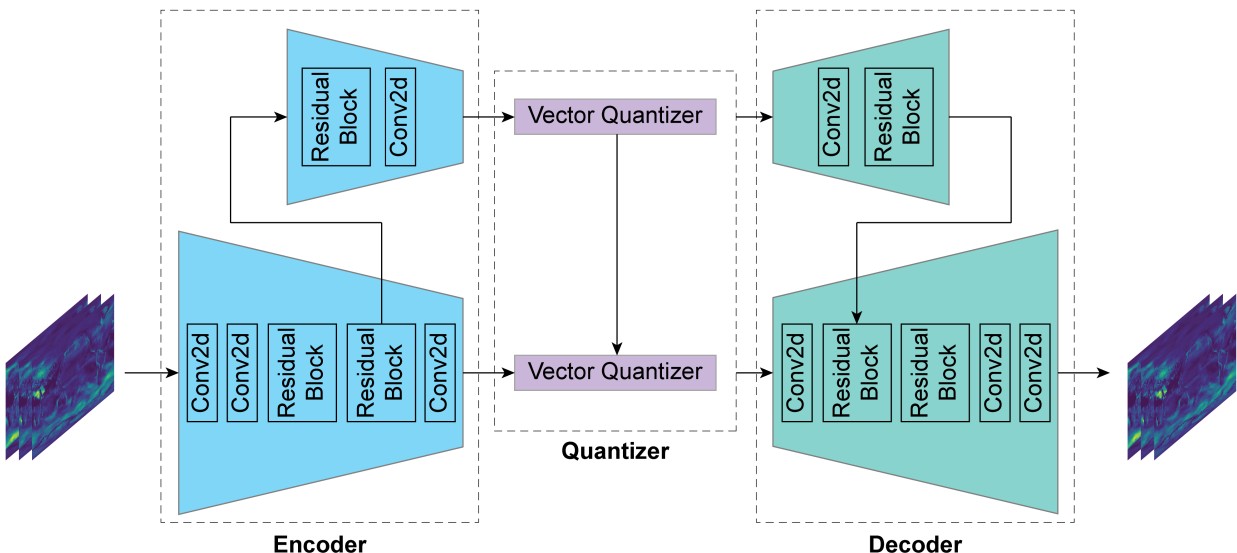

Figure 1: Model Architecture

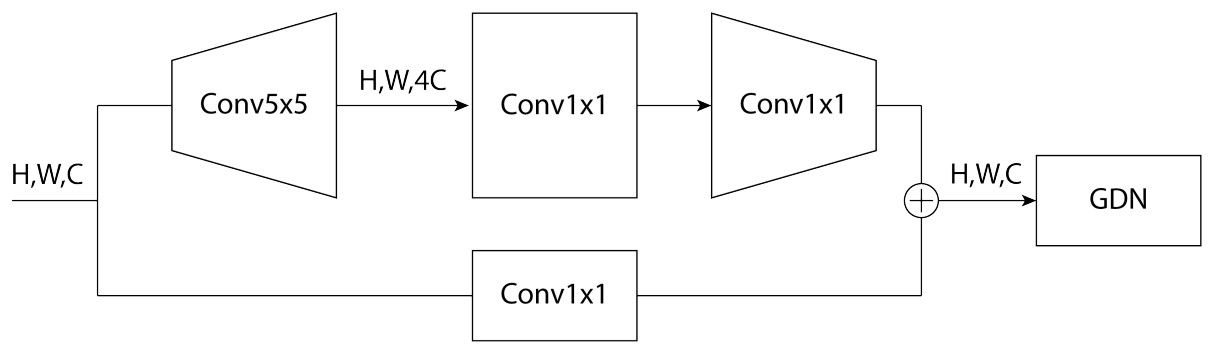

Figure 2: Components of a residual block

Table 1: The implementation of the model architecture

| Network | Stage | Operator | Stride | #Channels | #Layers |
|---|---|---|---|---|---|
| Encoder | Norm | N/A | N/A | N/A | 1 |
| | PreBlock | Conv4x4 | 1 | 32 | 2 |
| | EncRes_0 | EncRes | 2 | 64 | 3 |
| | EncRes_1 | EncRes | 2 | 128 | 3 |
| Decoder | DecRes_1 | DecRes | 2 | 64 | 3 |
| | DecRes_0 | DecRes | 2 | 32 | 3 |
| | PostBlock | Conv4x4 | 1 | 1 | 2 |
| | De-norm | N/A | N/A | N/A | 1 |
| Quantizer | VQ_0 | Quantization | N/A | N/A | 1 |
| | VQ_1 | Quantization | N/A | N/A | 1 |

construction errors, mean-squared error (MSE) and multi-scale structural similarity (MS-SSIM). Depending on targeting criteria, either measure can be used to achieve desirable outcomes. Both measures have been shown to be good metrics since they generally output generated images with high quality (Minnen & Singh, 2020; Ballé et al., 2016; 2018b).

The encoder network E is created hierarchically. The first level aggressively reduces the dimension of the inputs and learns their data representation. The second level also performs slight dimension reduction. Data representation from the second level is quantized by its corresponding vector quantizer. The quantized values are then fed into the first-level vector quantizer. The second-level quantization acts as a prior to the first-level quantization. The additional information from the second-level quantization improves the capability of the first-level quantization, which leads to a better reconstruction quality. Even though the second level creates slightly more bits during compression, reconstruction quality improvement significantly outweighs a slight decrease in compression ratio.

The network E comprises several 2D convolution layers and blocks of residual connections. The first two convolution layers map inputs to a higher number of channels using a kernel size of 4. It is followed by a couple of residual blocks, which consist of strided convolutions with a kernel size of 5. Components of a residual block are illustrated in Figure 2. The residual block construction is inspired by the model proposed by (Tan & Le, 2021), which achieves high performance for classification tasks without significantly increasing the model size. Moreover, residual blocks alleviate the vanishing gradient problem and enable the implementation of deeper models, which improves the expressiveness of our networks (He et al., 2016). We use non-linear GELU functions as our activation functions (Hendrycks & Gimpel, 2016). A generalized divisive normalization (GDN) is used to normalize residual block's outputs and transform their distribution to be more Gaussian (Ballé et al., 2015). GDN is shown to be effective for both image compression (Ballé, 2018) as well as scientific data compression (Liu et al., 2021a). The encoder network can be simply represented as a mapping function as shown in equation 1.

$$z_e = \mathbf{E}(x) \tag{1}$$

The decoder network D is a mirror of the encoder network E. Transposed convolution layers are used to replace strided convolutions. Transposed convolutions at the beginning of each hierarchy alter decoder inputs to acquire suitable representation with C channels for the following residual blocks. In general, all blocks of the decoder loosely reverse all operations performed by the encoder. Network D maps the latent representation back to the original dimension, outputting reconstructed data. The decoder can also be considered to be a mapping function as shown in equation 2.

$$\hat{x} = \mathbf{D}(\text{decoder\_inputs}) \tag{2}$$

### 3.2 Vector Quantizer

Although vanilla AE can perform dimension reduction, it cannot flexibly generate data given fixed inputs. Variational Autoencoder (VAE) (Kingma & Welling, 2013) and its variants are implemented to improve reconstruction performance. VAEs not only minimize the reconstruction loss but also learn the distribution of the latent variable by optimizing the Kullback–Leibler (KL) divergence. As a result, a more diverse set of images can be generated with much higher quality (Vahdat & Kautz, 2020; Sønderby et al., 2016).

Based on the idea of VAE, we impose slightly different criteria on the objective function. Following a proposed approach implemented in Vector Quantized Variational Autoencoder (VQ-VAE) (Razavi et al., 2019), our model is trained to minimize the reconstructed loss, i.e. L2 distance, as well as optimize discrete codebooks. The latent representation encoded by the encoder is projected onto codebook vectors. The vector, which has the smallest Euclidean distance to the encoded latent variable, is selected to become the decoder's inputs as shown in equation 3

$$z_q = \mathbf{Q}(z_e) = \text{argmin}_{q_k \in Q}(||z_e - q_k||) \tag{3}$$

The quantizer outputs a set of integer values, which are indices of quantized vectors. These indices are then further compressed using a lossless compression scheme, i.e. Huffman coding-based algorithms. The size of compressed quantized data is significantly reduced because quantized values are in the form of integers, which are efficiently compressed by any lossless compression algorithm.

Our training procedure for our codebooks is similar to the method described in (Razavi et al., 2019). Each codebook in the model is updated using an exponential moving average with a decay of 0.99. The update is measured based on changes in codebook entries after each training iteration. A straight-through estimator (Bengio et al., 2013) is implemented to overcome the discontinuity of gradients created by discrete codebooks. The estimator acts as an identity function that identically maps the gradients of the decoder to the encoder in the backward propagation.

### 3.3 Preprocessing Large-scale Data

### 3.3.1 Data Standardization

In this work, we focus on compressing large-scale high-resolution scientific data obtained from Earth's simulation. Since each set of data has its own data distribution, it is important to preprocess raw data prior to training. Statistical measures of data can be analyzed based on each specific data type. The availability of statistics enables us to use Gaussian standardization for data whose distribution is Gaussian. The technique is also applicable to distribution that approaches the Gaussian distribution. The standardization method is shown in equation 4.

$$x_{st} = \frac{x - \mu}{\sigma} \tag{4}$$

where $x$ is a data value, $\mu$ is the mean of the data, and $\sigma$ is the data standard deviation.

The inverse of standardization is required for converting the reconstructed data back to the actual value range. The inverse is formulated in equation 5.

$$x = \mu + x_{st} * \sigma \tag{5}$$

However, if the data distribution is not Gaussian, directly applying standardization does not improve compression performance. In this scenario, logarithm scaling is a technique to transform the original data to its corresponding logarithmic scale. The technique usually changes the data distribution to be close to Gaussian, which enables us to effectively use the standardization method on the data.

### 3.3.2 Missing Value Handling

Data masking is necessary for data compression in many cases. In many scientific simulations, there are regions that are not of interest to the researchers conducting experiments. Those areas are generally assigned values that are extremely negative or easily distinguished from actual simulation values. Therefore, we use masking layers to indicate valuable values and ignore unwanted regions in our model. Even though the masking increases the storage size, this redundancy is negligible since it is made up of several integer values, which can be significantly compressed by any standard lossless compression algorithm such as Huffman coding-based compression schemes.

Missing values in data are also replaced by a different value. The replacing values can be the mean or the median of the entire available data. For simplicity, we assign missing values with the data mean since data statistics are readily available. After cleansing missing values and masking the data, the data and their corresponding masks are partitioned into small blocks.

### 3.3.3 Data Partitioning

Machine learning models generally cannot handle raw scientific data, since each dimension of any data is large, which cannot fit into the system's memory. To overcome this issue, data are partitioned into small blocks prior to training or compression. Each dimension of a block is a power of two. Particularly, we restrict the block to having a height and width of 64 for the training process, as we observe that this setting achieves the best reconstruction quality. Moreover, a power of two in each block dimension makes the up-sampling

and down-sampling efficient. No padding or trimming is required for the outputs, which saves additional computing power.

However, the shapes of raw data are not always divisible by two, which is a requirement to have a block size of a power of 2. Then, data whose size is not a multiple of block size are padded. Padding is performed at the edges of each dimension. For Earth's simulation data, we cyclically replicate data values at one edge and concatenate them at the other end. For example, to pad the left edge of 2D data, values on the right edge are copied and appended to the opposite side. This padding pattern is especially helpful for treating continuous simulation data with periodical boundary conditions, e.g., climate modeling data.

The partitioning technique mentioned above works well in general. However, as all partitioned blocks are discrete, the whole set of partitions does not include any transition from one block to its adjacent neighbors. To smooth out the boundary and make the transition from one block to another more accurate, an overlapping block partition technique is implemented (Kim et al., 2022). Instead of making mutually exclusive blocks of data, adjacent blocks are partitioned in a way that they overlap with each other in a small area. In particular, assuming each block is of size 64 and there is an overlap of eight, the second block is created to contain the last eight values of the first block as well as the next 56 values. The data overlapping technique is only implemented for training data, whereas the discrete data partitioning technique without overlapping is used for testing and compression.

### 3.4 Objective Function

#### 3.4.1 Reconstruction Loss

The reconstruction loss is the discrepancy between the reconstructed and original data. We minimize the L2 distance of the target and compressed data, i.e. $l_{recon}(x, \hat{x}) = ||x - \hat{x}||_2$. The minimization simply matches the compressed data to the original data as closely as possible.

#### 3.4.2 VQ commitment loss

The commitment loss accounts for the difference between the quantized codebook vectors and outputs of the encoder. Since quantization distorts the data, decreasing the distance between the quantized vectors and the original data reduces the distortion. We impose an L2 distance constraint on the codebook vectors and their corresponding inputs. The commitment loss, $l_q$, is defined as in equation 6.

$$l_q(z_e, z_q) = ||z_e - z_q||_2 = ||z_e - Q(z_e)||_2 \tag{6}$$

Where $z_e$ and $z_q$ are outputs of the encoder and their corresponding quantization values, respectively.

Overall, the model is trained to optimize the following objective

$$L = \lambda_{recon} * mask * l_{recon} + \lambda_q * l_q \tag{7}$$

where $mask$ is a masking layer, which indicates which data points should be taken into account in optimization; $\lambda_{recon}$ and $\lambda_q$ are constant coefficients of the reconstruction and commitment losses, respectively. The constant $\lambda_q$ is set to be 0.25 following the suggestion by (Van Den Oord et al., 2017). Meanwhile, $\lambda_{recon}$ is set to 2 because the parameter slightly affects the trade-off between reconstruction quality and compression ratio. The objective function in equation 7 is acquired based on the assumption that quantization values are uniformly distributed. Uniform distribution leads to a removal of an additional KL term in the objective because the term becomes a constant with respect to encoder parameters (Van Den Oord et al., 2017).

### 3.5 Error-bounded Technique

Reconstructed data from neural networks sometimes have large distortions from the original data. To counteract the large distortion of some reconstructed values, a straight-through technique is introduced. The

Table 2: Basic information of benchmark data from SDRBench

| Dataset | Dimension | Domain | Field |
|---------|-----------|--------|-------|
| CESM 2D | 1800x3600 | Weather | CLDHGH CLDMED CLDLOW CLDTOT |
| CESM 3D | 26x1800x3600 | Weather | CLOUD |
| NYX 3D | 512x512x512 | Cosmology | Temperature Baryon density |

Table 3: Basic information of SST data

| Data size | Dimension | Lowest value | Highest value |
|-----------|-----------|--------------|---------------|
| 111.90 GB | 3240x1800x3600 | -1.95 °C | 34.84 °C |

straight-though technique classifies reconstructed values into two groups, predictable and unpredictable. Reconstructed data that meet the tolerance constraints are called predictable values. In other words, predictable data have error values less than or equal to a predefined threshold. Otherwise, they are unpredictable values. Unlike predictable values, which can be used directly as final reconstructed values, unpredictable values have errors that exceed the threshold. Thus, corresponding true values and their locations are saved separately on a file to replace unpredictable values during reconstruction.

## 4 Experiments

### 4.1 Resource Availability

This paper uses existing, publicly available data from SDRBench (`https://sdrbench.github.io/`) for bench-marking the performance of our model. As for the compression on our application data, the model compresses the High-Resolution Earth System Prediction (iHESP) data. The iHESP data have been deposited at `https://ihesp.github.io/archive/` and are publicly available.

### 4.2 Benchmark Data: SDRBench

Our proposed models are initially tested on published scientific benchmark data, SDRBench (Zhao et al., 2020). This benchmark provides numerous simulation data from many different fields, ranging from electronic structures of atoms, molecules to weather, and cosmology. The benchmark is publicly available for different scientific purposes.

Even though we focus on compression 2D data, a couple of 3D data sets are also being compressed to verify the possibility of generalizing our architecture to higher-dimension data. Table 2 summarizes several data sets and some fields we use for our compression.

The 3D CESM data include comprehensive attributes of cloud properties for many different altitudes, which can be viewed as many 2D data stacking on top of each other. Therefore, we use the 3D CESM data as a training set for CESM cloud data, whereas all snapshots of 2D CESM data are our testing data.

### 4.3 High-Resolution Earth System Prediction (iHESP) Data

The International Laboratory for High-Resolution Earth System Prediction (iHESP)(Chang et al., 2020) was a project aiming to develop more advanced modeling frameworks for high-resolution multiscale Earth system predictions to improve the simulation and prediction of future changes in extreme events. iHESP

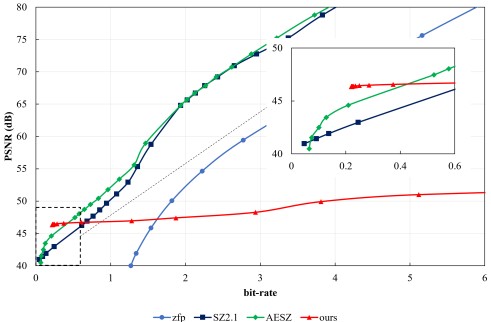

Figure 3: Compression performance on CESM 2D CLDHGH data

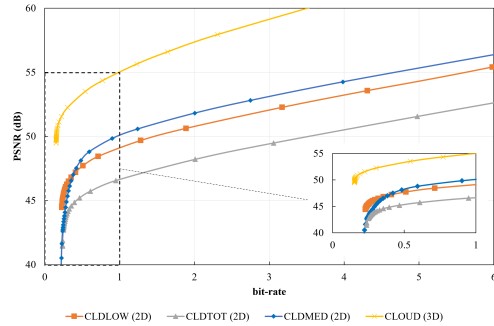

Figure 4: Compression performance of the proposed model on different CESM cloud data

also provides numerous global and regional high-resolution simulation data spanning hundreds of years. The global climate was simulated using different high-resolution configurations of CESM version 1.3 for atmosphere, land, ocean, and sea-ice. Meanwhile, regional data were generated from the ocean model ROMS (Regional Ocean Modelling System) with the atmospheric model WRF (Weather Research and Forecast model) using the CESM/CIME coupling infrastructure. All data are also publicly accessible.

Among a large array of ocean properties provided by iHESP, sea surface temperature (SST) is one of the most important attributes for the ocean. The property is simulated over hundred years, which leads to a substantial amount of storage needed to store the data. However, the large amount of available data also enables us to leverage machine learning for compression.

Basic information of SST data is presented in Table 3. The first dimension of the data represents the time evolution. The next two dimensions are the height and width of the data, respectively. General ocean information, such as simulation history and climate coefficients, are also included in the metadata of the data set. Latitudes and longitudes are also available to scale the data back to the global coordinate system when it is required.

Data preprocessing is crucial for SST in both training and compression. Temperature values are only available where sea water presents, whereas undefined values are assigned to continents. In other to deal with missing values, a masking layer is created to differentiate between those regions.

The data are split into two sets, a training set and a testing set. The training set contains almost ∼100GB of SST data while the testing set consists of temperature data of the last 120 consecutive months in the simulation. Data in the training set are partitioned using the overlapping technique, while we apply the discrete partitioning technique for the testing set. Both training and testing sets contain blocks of data of size 64. During compression, data are partitioned into blocks of size 256 for better resolution.

## 5 Results and Discussion

### 5.1 Compression of Benchmark Data

#### 5.1.1 2D Data

The compression performance of our models on different data sets is compared to other compression models, namely SZ2.1 (Liang et al., 2018), ZFP (Lindstrom, 2014), and AESZ (Liu et al., 2021a). Figure 3 shows that our proposed model outperforms other compression schemes when bit-rates are below 0.40, which are equivalent to compression ratios of greater than 80. At a very low bit-rate of 0.22, the reconstructed data of our model has a PSNR of 46.35 dB. This is an improvement from the hybrid AESZ model, which requires a bit-rate of around 0.41 to obtain the same PSNR.

However, the PSNR of the proposed model does not follow the same trend as the compression performance of other compression models. Our trained model has a fixed set of parameters. To increase PSNR without

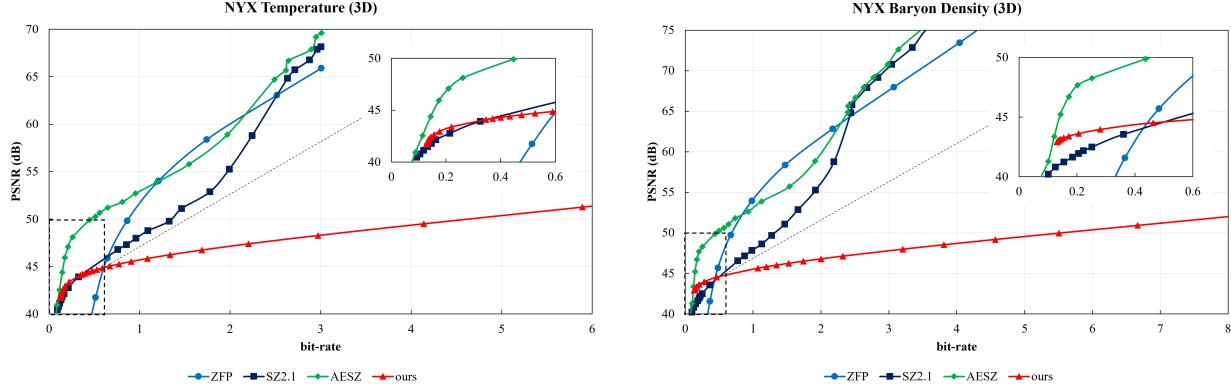

Figure 5: Compression performance on NYX 3D data. Left: Compression on NYX temperature. Right: Compression on NYX Baryon density

training a different model, we apply the straight-through method to restrict the error-bound of the reconstructed data. There is a possibility of training different models with larger latent variables and codebooks to get much higher PSNR at any certain bit-rate. However, with the goal of targeting the low bit-rate regime, exhaustively exploring all possible combinations of neural networks over a wide range of bit-rates is not in the scope of this work.

Our compression model is also leveraged to compress different 2D data. Compression performance on many different cloud data is illustrated in Figure 4. Since the CESM 3D CLOUD data should be treated as 2D data as suggested by domain scientists (Zhao et al., 2020), its compression results are presented together with other 2D data. It is worth mentioning that compression on all CESM cloud data uses the same model architecture with the exact same weights. Even when applying the model to these data, it obtains high PSNR while maintaining a very low bit-rate. The compression performance indicates that this particular model for CESM cloud data achieves a good generalization.

### 5.1.2 3D Data

The proposed model achieves reasonable compression on 3D benchmark data. As can be seen from Figure 5, at low bit-rates, our model surpasses SZ2.1 and ZFP in terms of performance. However, the quality of reconstruction from the hybrid model - AESZ - is higher than our model. One of many possible reasons for the weaker performance of our compressor is that our model is designed using primarily 2D convolution layers. Therefore, it does not have the extensive capability to learn data representation in 3D. On the other hand, when compressing 3D data, AESZ changes its machine learning architecture to 3D convolution neural networks. This change is one of the factors that boost the compression performance for volumetric data.

### 5.2 Compression of iHESP Sea Surface Temperature (SST) Data

Compression results for the testing set of high-resolution SST data show that the model can reconstruct data with high quality while maintaining a high compression ratio even for large-scale simulation data. As can be seen from Figure 6, after being compressed by a factor of 240, the reconstruction achieves a PSNR of 50.16. Moreover, in terms of visualization, it is unlikely to detect differences between the original and reconstructed data. However, there are some slightly noticeable distortion areas, especially along coastal lines between oceans and continents. Since data are only available for sea water, data points on continents are set to be a suitable constant. The assignment of the constant creates large variations in values along the edges of continents, which hinders the reconstruction ability of the model in those particular regions.

Table 4 presents the compression performance of the model on the whole testing data. The quality of reconstruction, PSNR, of each snapshot varies from 48.58 to 51.5. The reason for the differences in PSNR is that data distribution of each snapshot differs from time to time, which leads to the variation in quantization

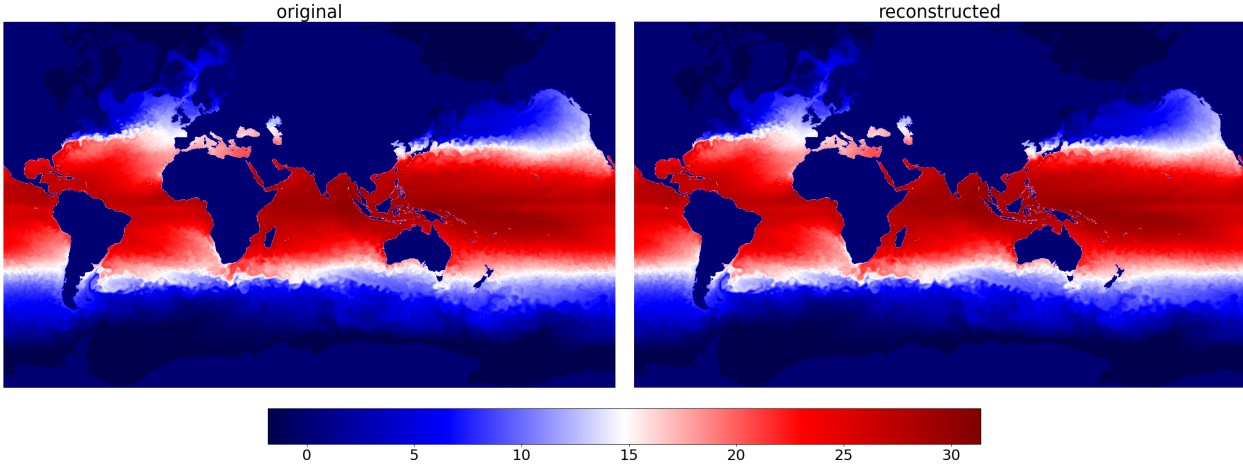

Figure 6: A snapshot of the sea surface temperature (SST) data (original value range: [-1.82°C; 31.38°C]; PSNR: 51.31)

Table 4: Compression performance of the proposed model on testing SST data (data size: 4,144.60 MB)

| Metrics | Results |
| --- | --- |
| Compression ratio | 231.54 |
| PSNR (dB) | 50.04 |
| Compression speed (MB/s) | 96.53 |
| Decompression speed (MB/s) | 87.43 |

values from codebooks; hence changes in the reconstruction quality. Nevertheless, the deviation of the snapshots' PSNR does not vary far from the average of 50.04, which indicates that our model achieves stable performance over all data sequences.

Compression and decompression speeds are also acceptable. Compression speeds on HPC nodes are presented in Table 4. On average, it takes around 45 seconds to complete either compression or decompression for 4GB data. On a personal computer with an NVIDIA 3060 Ti accelerator, compression and decompression both take around one and a half minutes on the same data. The small difference in the two platforms indicates that the compression pipeline is primarily bottlenecked by the data transferring between CPUs and GPUs. However, compression speed on the personal computer shows promising results that the model is also suitable for compression on small devices.

### 5.3 Ablation Study on iHESP Sea Surface Temperature (SST) Data

We conduct experiments with different levels of quantization to obtain the best trade-off between compression ratio and reconstruction quality. Generally, increasing the number of quantization layers should improve the quality of the reconstruction while reducing the compression ratio. However, as shown in Table 5, our results show that the two-stage architecture achieves the lowest MSE on the testing sea surface temperature (SST) data. Although our implementation can scale up to an arbitrary number of quantization layers, the construction quality does not always improve. One possible explanation for this phenomenon is that when the architecture becomes too deep, it might over-fit the training data, which leads to the worse performance.

In terms of data preprocessing techniques, using the two-stage model, the combination of uniform masking and overlapping partitioning method achieves the best performance over other techniques (Table 6). For the weighted masking model, we use larger weights for regions, which are more important during simulation,

Table 5: Compression performance of different model architectures on SST data

| Model | Testing MSE |
|---|---|
| Single Stage Quantization | 0.119 |
| Two Stage Quantization | 0.012 |
| Three Stage Quantization | 0.030 |

Table 6: Compression performance of the two-stage quantization model for different preprocessing techniques on SST data

| Model | Testing MSE |
|---|---|
| masking + overlapping partition | 0.012 |
| weighted masking + overlapping partition | 0.021 |
| masking + discrete partition | 0.020 |

while lower the weights for other sections. However, the performance of the weighted version reduces the testing MSE since the model focuses more on those targeted regions with the compromise for other areas.

### 5.4 Limitation

One of many limitations of the proposed model, and possibly the biggest limitation, is the training of neural networks (NNs). NNs perform best on data which have similar distribution with the data being trained on. Therefore, if out-of-distribution data are given to a machine learning model, the results might not be reliable and, oftentimes, incorrect. In our case, which is the Earth system, each distinct attribute requires a different model. This limitation can limit the usability of our proposed model. However, transfer learning techniques can be used to reuse the architecture for different types of data (Wang et al., 2022).

Secondly, one architecture might not perform well for different types of data because of the difference in data distributions. Each distribution has a different property which makes it difficult to select the correct architecture for that data. As a result, exploring an optimal architecture for a data type might require tremendous effort.

## 6 Conclusions

Our proposed model shows to be effective in compressing floating-point scientific data, both on 2D benchmark data and our large-scale high-resolution data. It achieves a high compression ratio while preserving a high quality of reconstruction. The model outperforms other state-of-the-art models in some benchmark data sets, particularly 2D simulation data. However, there is room for further improvement. Other lossless compression schemes, such as arithmetic coding, which offers better compression performance, can be used to replace Huffman coding. The model can also be further improved by optimizing the rate loss term, which potentially leads to a better compression ratio. Furthermore, the compression pipeline of the proposed model can be optimized to improve the speed of compression. Since scientific data compression using neural networks is still in its early age, there is so much more potential improvement that can be achieved for future research along this line.

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

## A  Additional experiments

### A.1  Frequency Loss Term

We conduct experiments which take into account loss terms in the frequency domain of the data. Both inputs and reconstruction are transformed into the frequency domain using Fast Fourier Transform (FFT). The L2 distance between the two transformed sets are then calculated using equation 8. This added term forces the model to directly minimize errors between high and low frequency components of the L2 distance.

$$l_{fft}(x, \hat{x}) = ||\text{FFT}(x) - \text{FFT}(\hat{x})||_2 \tag{8}$$

Where $x$ and $\hat{x}$ are inputs and reconstructed data, respectively.

The FFT loss, $l_{fft}$, is added with other losses to create an objective function of the model as shown in equation 9.

$$L = \lambda_{recon} * mask * l_{recon} + \lambda_q * l_q + \lambda_{fft} * l_{fft} \tag{9}$$

where $\lambda_{recon}$, $\lambda_q$, and $\lambda_{fft}$ are constant coefficients of the reconstruction, commitment losses, and FFT loss, respectively.

## A.2   Results

Our model trained with the added FFT loss performs reasonably well for the iHESP sea surface temperature (SST) data set. At a compression ratio of 221.63, the model achieves a PSNR of 47.04 for the reconstruction. Despite of having a good quality of reconstruction, its performance is surpassed by the model trained without the added FFT loss as discuss in section 5, which achieves an average PSNR of 50.04 at a compression ratio of 231.54. One possible explanation for the lower reconstruction quality is that there is a trade-off between the MSE terms in the time domain and the frequency domain during training. While the MSE loss term in the time domain learn data representation in a particular region, the FFT loss term focuses on different regions. As a result, the quantitative result, PSNR, of the "FFT model" is outperformed by its counterpart.

## B   Additional Visualization Results

Additional results of the compression using our proposed model are provided in this section. Figure 7 - 13 show visualization results for compression on different data sets.

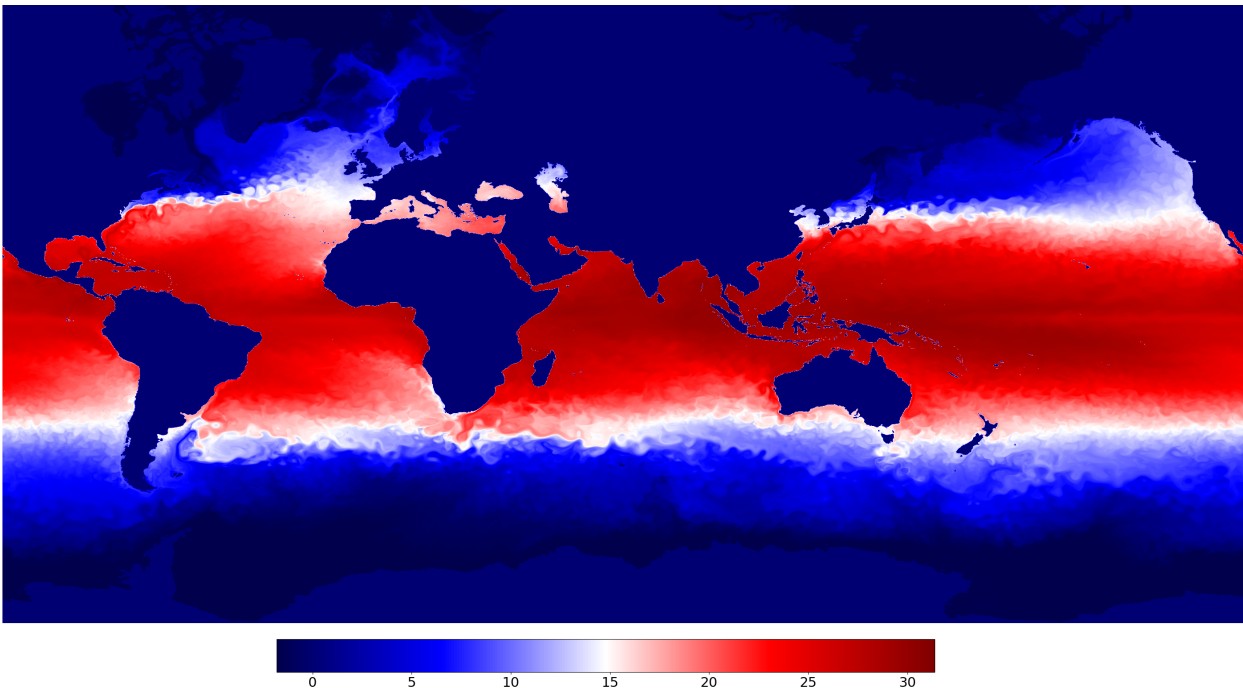

Figure 7: A snapshot of the sea surface temperature (SST) original data

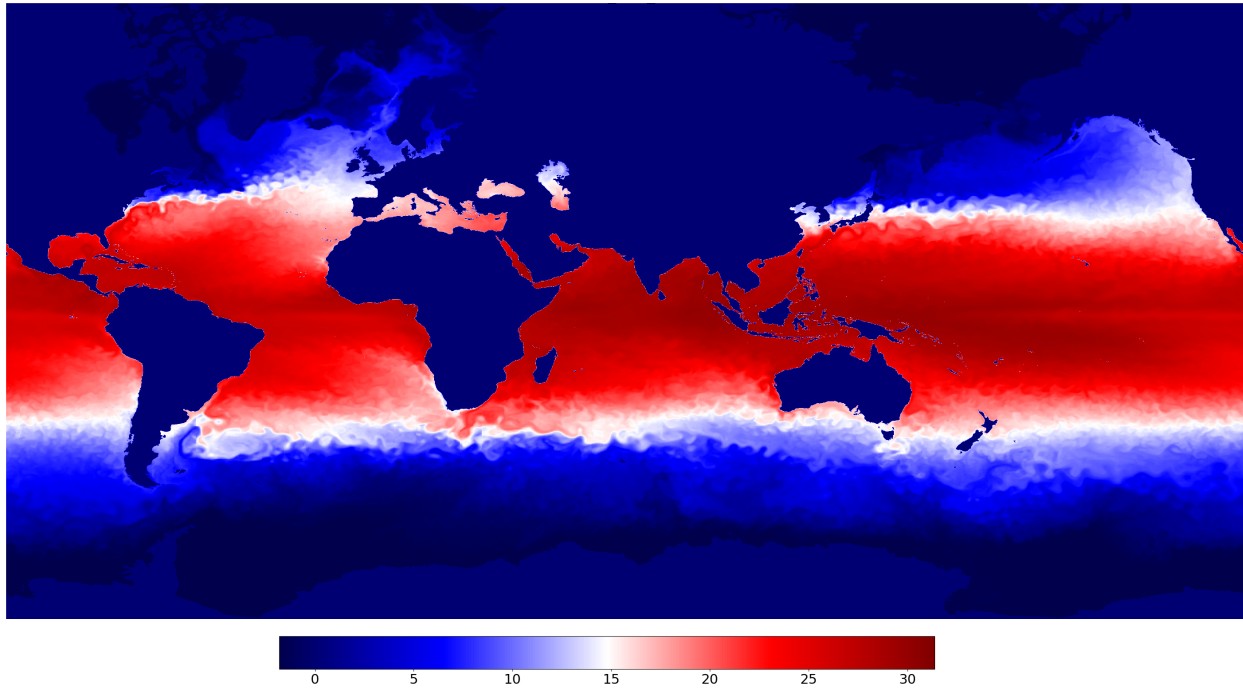

Figure 8: A snapshot of the sea surface temperature (SST) reconstructed data of the data in Figure 7 (PSNR: 51.31, compression ratio: 231.54)

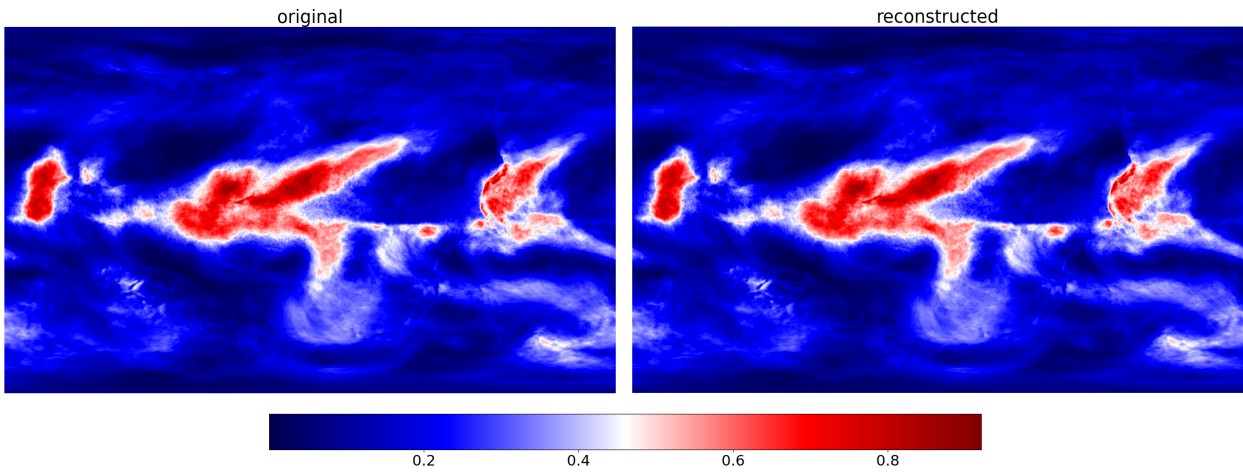

Figure 9: Compression for CESM CLDHGH 2D data (original value range: [3.38e-07;0.92]; compression ratio: 122.73; PSNR: 46.35)

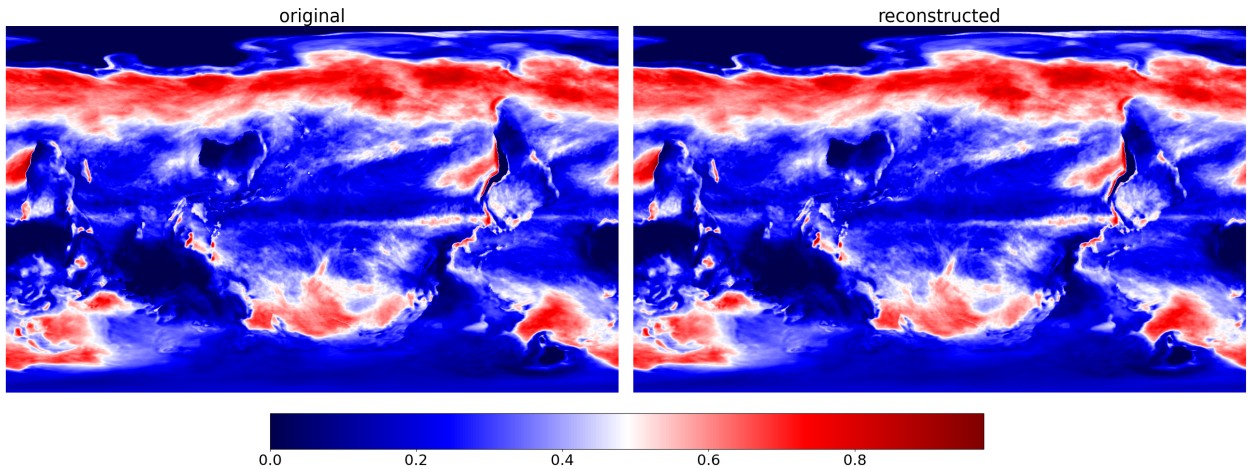

Figure 10: Compression for CESM CLDLOW 2D data (PSNR: 44.87, compression ratio: 136.41)

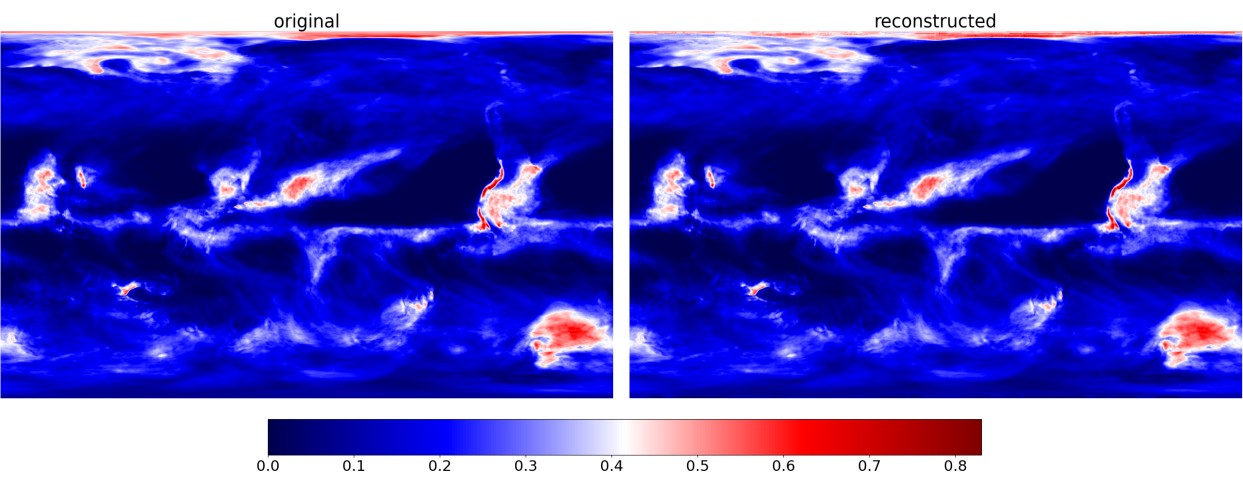

Figure 11: Compression for CESM CLDMED 2D data (PSNR: 42.63, compression ratio: 133.99)

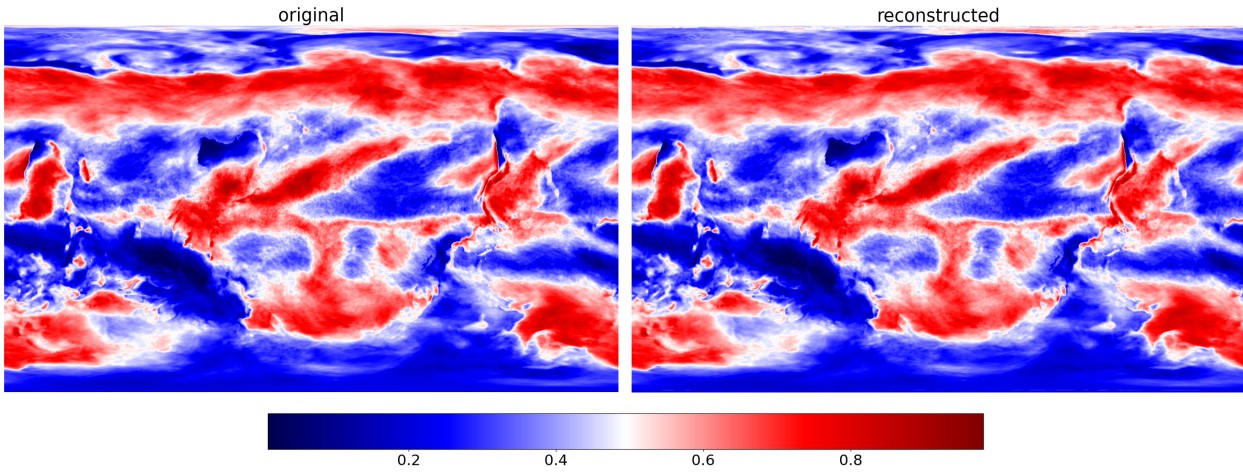

Figure 12: Compression for CESM CLDTOT 2D data (PSNR: 42.71, compression ratio: 127.87)

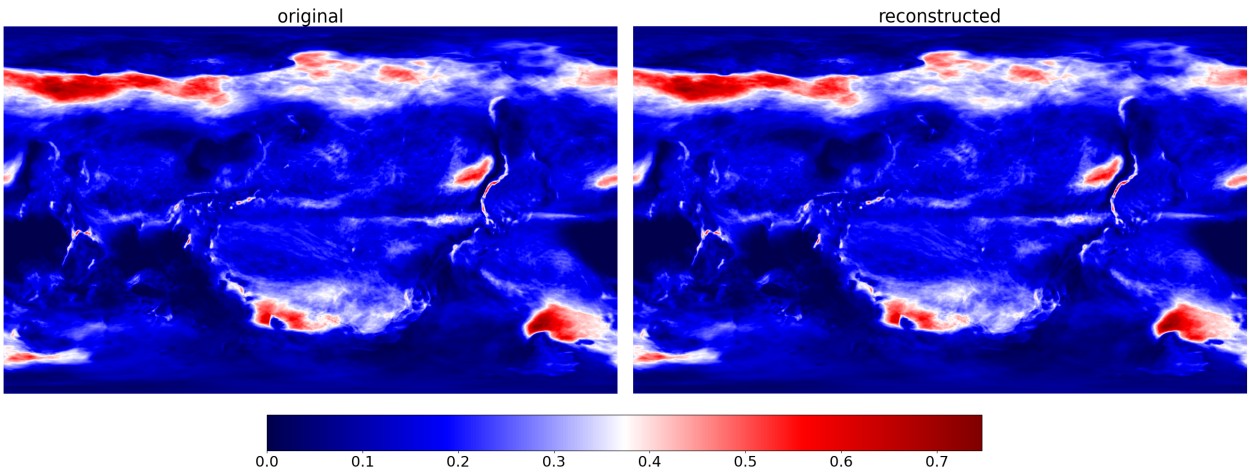

Figure 13: Compression for a snapshot of CESM CLOUD 3D data (PSNR: 48.51, compression ratio: 210.97)

