# OpenReview forum: "Hierarchical Autoencoder-based Lossy Compression for Large-scale High-resolution Scientific Data"
_TMLR — Rejected by TMLR_

### Review · Reviewer_SYv9 · 2023-11-06

**Summary Of Contributions:**

This paper proposed a compression framework for scientific data. Some preprocessing and training strategies have been proposed to improve the performance. Experiments show good performance compared with some existing methods at low bitrate levels. Multiple scientific datasets are tested including 2D and 3D datasets.

**Audience:**

Yes

**Claims And Evidence:**

Yes

**Requested Changes:**

1. The method used for the prediction between the two-stage quantization is not clearly staged and should be included.
2. Some details may be missing including the codebook size, training time, and computational complexity.
3. The rd curve provided in Figure.3 of the proposed method is kind of weird, with minimal quality gain when the bitrate increases. Also, the lowest bitrate is not low enough to better compare with other methods.
4. The performance of more models trained to cover a relatively larger bitrate range should be provided.

**Strengths And Weaknesses:**

**Strengths:**

This paper provided a framework for scientific data compression. The performance at low bitrate is good. Different datasets including 2D and 3D data are tested.

**Weaknesses:**

1. The overall framework is simple. Most of the components are from existing works, with marginal performance gain. The experiment part can also be improved with more analyses.
2. The authors mentioned that the overlapped partitioning is only for training. However, from the experience in VAE-based compression frameworks, there are indeed some tricks that overlapped partitioning only during tests can provide better performance, which is inconsistent with the conclusion from this paper.

---

### Review · Reviewer_fi25 · 2023-11-10

**Summary Of Contributions:**

The authors propose a vector-quantized variational autoencoder (VQ-VAE) architecture to compress scientific data lossily, focusing on simulated climate data in particular.

**Audience:**

No

**Claims And Evidence:**

No

**Requested Changes:**

Unfortunately, I am not convinced that the authors can address most of my concerns within the review window. The main thing the authors need to address is how their contribution is relevant to the TMLR community, present more convincing motivation for their method and significantly improve the writing.

**Strengths And Weaknesses:**

# Strengths
The authors' compression algorithm has better rate-distortion performance compared to baseline methods on low-bitrate regimes.

However, as my area of expertise is not scientific data compression, I cannot verify that the baselines are sensible.

# Weaknesses
The authors' sole contribution is that they propose a VQ-VAE architecture for lossy compression that outperforms some previous methods on some scientific datasets.

However, the authors do not make any new methodological contributions at any level. In terms of model class, they use a standard VQ-VAE. As for the architecture, the authors' justification for their choice seems to be that it works. Their proposed architecture does not seem to be inspired or based on domain-specific insights, nor does it appear to result from a rigorous selection process.

It is also unclear whether the regime where the author's method works best is useful. In particular, the authors claim that "the reconstruction error is negligible for scientific analysis." However, they don't define or reference any source defining negligible error for the data they are considering.

The authors' method is also not replicable. In particular, the description of the data is lacking, which also means it is unclear how they pre-process the data. For example, the authors note that they perform a log-transform on the data: "In this scenario, logarithm scaling is a technique to transform the original data to its corresponding logarithmic scale. The technique usually changes the data distribution to be close to Gaussian, which enables us to effectively use the standardization method on the data." Besides being theoretically problematic, it is unclear how this transform applies to, e.g., the climate data the authors consider, as temperatures can be negative, so the logarithm cannot be applied to them.

Furthermore, the authors describe another component of their pipeline, which they call a "straight-through technique." However, the authors provide no details about this component.

I am also concerned about how the compression rate is calculated: the authors do not include the parameters of their model in the rate. This is fine for common data modalities such as images, as a single model can generalize over many different image distributions. However, I imagine that an autoencoder trained on a particular type of simulated climate data can only efficiently compress that particular type of climate data, so the model should always be saved alongside the compressed data so that it can be decoded.

Also, how do the authors deal with the data being on a sphere? The authors' padding method of wrapping around the edges implies that the earth is topologically equivalent to a torus, not a sphere.

Finally, the writing is of low quality at each level. Most of the related works section is about works that are not related. For example, the authors discuss autoencoder-based image, video, and audio compression methods, which are of no direct relevance to the authors' choice of model class or architecture. In particular, the authors do not describe how any of the works mentioned in the section are related to their work.

Furthermore, many individual sentences are unclear, e.g.:
 - "Furthermore, recent studies by (Baker et al., 2014), (Sasaki et al., 2015), and (Baker et al., 2017) showed that reconstruction data from lossy compression can be used for post-hoc analyses" - what type of post-hoc analyes?
- "Benchmark results are also presented to ensure the performance of our model"
- "since the quantization layer of their compression models prevented the gradients from flowing through the networks, independently and identically distributed uniform noise was used to replace the quantization layer during training."
- "Transform compression (e.g. ZFP (Lindstrom, 2014)) transformed the data before applying other techniques, e.g. embedded coding, to truncate the transformed data. Coefficients and bit-planes determined by the model were used to decompress data. Increasing the number of coefficients and bit-planes improved the quality of reconstructed data but decreased the compression ratio." - What coefficients? What are bit-planes?
- "Machine learning models generally cannot handle raw scientific data, since each dimension of any data is large, which cannot fit into the system's memory." - What do the authors mean by each dimension being large?

Finally, there are more minor issues, such as:
- Figs 3-5: The label font size is too small; please increase it
- Fig 6: Please include a plot of the residuals. It is very difficult to tell the differences by eye.
- The citations are formatted incorrectly. Please use `\citet` for in-text citations and cite the labs and observatories correctly.
- There are several undefined acronyms in the text, such as SZ, ZFP, AESZ, CESM.

---

### Review · Reviewer_EHs8 · 2023-12-21

**Summary Of Contributions:**

This paper presents a neural network-based method for scientific 2D and 3D data compression. Based on existing works of learned image compression and vector-quantization (VQ) VAE, the authors demonstrate an architecture combining these two that can be used to learn compression model for scientific data. Experimental results show that a low-bit-rate region, the proposed method has better reconstruction quality than existing ones.

**Audience:**

Yes

**Broader Impact Concerns:**

There is no ethical concern in this paper.

**Claims And Evidence:**

No

**Requested Changes:**

To get this paper accepted, I think the authors should try to revise the related work and rephrase their design motivations for the weaknesses in technical soundness, and redesign the experiments to address the major weaknesses in experimental evaluation. It is also important to add back the necessary details in the description of their method and settings.

**Strengths And Weaknesses:**

### Strength:
Since there have been very few successful work to employ learned vector quantization for compression (primarily for images), it is good that the authors demonstrate that VQ can actually be used in a practical compression scheme and achieves better performance at lower bit-rates.

### Weakness:
**Technical Soundness**

My major concern for this manuscript is the technical soundness, specifically in terms of factual and logical correctness, terminology, technical rigor, and experimental evaluation validity.

- In the conclusion, the authors claimed that the proposed method outperforms other SOTA models. However, from the experiments shown in Fig. 3, for most of the bit-rate ranges the proposed one is underperforming others. Therefore, I don't think the claims are well supported by the experiments.
- The authors use "two-level quantization" to describe the scale hyperprior technique used in (Balle, 2018) and in their proposed method. From the image compression prospective, this terminology is inappropriate, unclear, and misleading. The core idea of hyperprior is to transmit a side information to provide more accurate probability estimation in entropy coding for main latent representation. The authors simply treat it as a "quantization", which is not the important aspect of hyperprior. Because of this, the authors' rationale for this technique is wrong. It does not make sense to me when the authors argue that the so-called second-level quantization "improves the capability" of the "first level quantization".
- It is discussed in the related work that when the neural network reduce the dimensionality by 8-fold, it theoretically compression the data by 8 times. This is not technically rigorous. The compression ratio depends on the dimensionality as well as the entropy. Even if the dimensionality is reduced, if the value range increased and therefore the entropy is likely increased, the compression ratio might be far higher than 8.
- I don't think it is rigorous to say that VAE is implemented from AE to "improve reconstruction performance".
- References are needed when the authors say "data masking is necessary for data compression in many cases".
- Bit-rates should have units. It makes no sense to just say the bit-rate is below "0.50".
- It is claimed that the proposed method outperform existing SOTA. However, Fig. 3 shows that the proposed method only outperform other methods in a very narrow range of low bit-rates. I wonder if at such low bit-rates the reconstructed data can serve scientific research purposes. Authors should justify that with more evidences.
- Fig. 4 compare the R-D performance of one single model on different data, which makes no sense to me. Different content definitely have different R-D curves. I don't think the conclusion the authors made can really be drawn from Fig. 4.
- Table 5 only presents MSE comparison, discarding the corresponding bit-rate. I don't think this comparison can be used to draw conclusion since the bit-rate would be different.

**Clarity**

I am also concerned about the presentation clarity, mainly in that some important information is missing.
- The authors didn't make it clear how their model achieves different rate-distortion points, which is very important for a lossy compression method. It is also not clear to me how they apply this model with 2D convolutions on 3D data.
- It is mentioned in the paper that if the data is unpredictable, the true values and locations are saved separately on a file. This is very confusing. If this is the case, these separate files also need to be counted to the total bits. The paper didn't specify what is the portion of these unpredictable and how they are handled.

**Organization**

The paper has a lot of missing details but spent a lot of room for unimportant descriptions. For example, Eq. (4) and (5) are not actually conveying valid information and should be removed.

---

### Decision · Action_Editor_cd9d · 2024-01-29

**Recommendation:** Reject

**Comment:**

While the study of neural network methods for scientific applications is of interest, all three reviewers raise concerns about both the technical and experimental sections. The authors have not provided responses to these comments. I believe these comments could be helpful for improving the manuscript, and the authors could consider incorporating them in the future.

**Audience:**

This work could be of interest to researchers working on data compression and the applications of deep learning models.

**Claims And Evidence:**

This paper studies neural networks for scientific data compression. The authors propose an architecture based on vector-quantization (VQ) VAE and demonstrate its performance as a  compression model for several 2d and 3d scientific data.

However, concerns have been raised by reviewers about the choice of the model architecture (VQ-VAE) for the domain-specific application, the validity of the experimental evaluation, and other factors related to the current manuscript. For instance, there are not sufficient experimental results to support the claim that the proposed method outperforms other SOTA models.